# Promoting electricity conservation through behavior change: A study protocol for a web-based multiple-arm parallel randomized controlled trial

**Mojtaba Habibi Asgarabad**[1]*, **Stepan Vesely**[1], **Mehmet Efe Biresselioglu**[2], **Federica Caffaro**[3], **Giuseppe Carrus**[3], **Muhittin Hakan Demir**[4], **Benjamin Kirchler**[5], **Andrea Kollmann**[5], **Chiara Massullo**[3], **Lorenza Tiberio**[3], **Christian A. Klöckner**[1]

1 Department of Psychology, Norwegian University of Science and Technology, Trondheim, Norway, 2 Sustainable Energy Division, Izmir University of Economics, Izmir, Türkiye, 3 Department of Education Science, Experimental Psychology Laboratory, Roma Tre University, Rome, Italy, 4 Department of Logistics Management, Izmir University of Economics, Izmir, Türkiye, 5 Energie Institut, Johannes Kepler University Linz, Linz, Austria

* Mojtaba.h.asgarabad@ntnu.no

**Data Availability Statement:** No datasets were generated or analysed during the current study. All relevant data from this study will be made available

## Abstract

### Background and aims

As a part of the framework of the EU-funded Energy efficiency through Behavior CHANge Transition (ENCHANT) project, the present paper intends to provide a "Research Protocol" of a web-based trial to: (i) assess the effectiveness of behavioral intervention strategies—either single or in combination—on electricity saving, and (ii) unravel the psychological factors contributing to intervention effectiveness in households across Europe.

### Methods and materials

Six distinct interventions (i.e., information provision, collective vs. individual message framing, social norms, consumption feedback, competitive elements, and commitment strategies) targeting electricity saving in households from six European countries (i.e., Austria, Germany, Italy, Norway, Romania, and Türkiye) are evaluated, with an initial expected samples of about 1500 households per country randomly assigned to 12 intervention groups and two control groups, and data is collected through an ad-hoc online platform. The primary outcome is the weekly electricity consumption normalized to the last seven days before measurement per person per household. Secondary outcomes are the peak consumption during the last day before measurement and the self-reported implementation of electricity saving behaviors (e.g., deicing the refrigerator). The underlying psychological factors expected to mediate and/or moderate the intervention effects on these outcomes are intentions to save electricity, perceived difficulty of saving energy, attitudes to electricity saving, electricity saving habit strength, social norms to save electricity, personal norms, collective efficacy, emotional reaction to electricity consumption, and national identity. The intervention effectiveness will be evaluated by comparing psychological factors and consumption

upon study completion either as a Supporting information file or to a stable, public repository.

**Funding:** The ENCHANT project is funded by European Union's Horizon 2020 research and innovation programme [project number 957115].

**Competing interests:** The authors have declared that no competing interests exist.

**Abbreviations:** ENCHANT, ENergy efficiency through Behavior CHANge Transition; GDPR, General Data Protection Regulation; ITT, Intent-To-Treat; MRA, Mixed Regression Analysis; NGOs, Non-governmental organization; NTNU, Norwegian University of Science and Technology; RCI, Reliable Change Index; RCT, Randomized Controlled Trial; SAGER, Sex and Gender Equity in Research; SCC, Standard Contractual Clauses; SPIRIT, Standard Protocol Items: Recommendations for Interventional Trials.

variables before and after the intervention, leading to a 14 (groups including 2 control groups) × 6 (time) mixed factorial design, with one factor between (group) and one factor within subjects (time)–6 measurements of the psychological factors and 6 readings of the electricity meters, which gives then 5 weeks of electricity consumption.

## Results

Data collection for the present RCT started in January 2023, and by October 2023 data collection will conclude.

## Discussion

Upon establishing feasibility and effectiveness, the outcomes of this study will assist policymakers, municipalities, NGOs, and other communal entities in identifying impactful interventions tailored to their unique circumstances and available resources. Researchers will benefit from a flexible, structured tool that allows the design, implementation and monitoring of complex interventions protocols. Crucially, the intervention participants will benefit from electricity saving strategies, fostering immediate effectiveness of the interventions in real-life contexts.

## Trial registration

This trial was preregistered in the Open Science Framework: https://osf.io/9vtn4.

# 1. Introduction

## 1.1. Background and rationale

While the commendable efforts of the European Commission to advance the climate neutrality by 2050 are acknowledged, a significant amount of work remains in the endeavor to curtail the considerable volume of $CO_2$ emissions stemming from energy generation. It is noteworthy that in Europe, households are responsible for 27% of primary energy consumption [1], emphasizing the pressing need to develop "electricity saving initiatives and intervention" that target this pivotal source of energy consumption and carbon emissions. Behavioral science has been making strides in designing interventions to enhance electricity saving by targeting key determinants influencing individuals' energy-related decisions, encompassing elements like communicating social norms, consumption feedback, informational provision, commitment goals, and competition [2–5]. However, a recent overview [6] indicates that research outcomes have been varied, predominantly leaning toward small or non-significant effects of these interventions on energy consumption.

The observed variance in outcomes can be attributed to two significant limitations in research on sustainable energy practices. First, upon putting into practice, the majority of behavioral interventions end up losing their effectiveness on electricity saving in real-life situations [7–9]. This ineffectiveness might be due to the evidence remaining limited in terms of scale of study, scope, and sample size (e.g., [10, 11]). Indeed, some of previous studies rely on small and homogeneous samples (e.g., one population segment in one country with particular cultural and psychosocial characteristics), which prevents them to access a broader demographic, thus limits the generalizability of findings to real-world scenarios [12, 13]. Addressing

the first limitation (scale, scope, and sample size) necessitates the execution of large-scale field experiments, capturing genuine Behaviors and contextual factors by studying choices made in real-life settings [14]. Consequently, an imminent requirement arises for overarching frameworks that ensure credibility and generalizability via empirical assessments and facilitate scalability through the implementation of expansive interventions [3]. The second limitation is that individuals' psychological factors can either facilitate or impede the desired outcomes of interventions, a facet that has received limited attention despite its theoretical underpinnings in economics and social sciences (e.g., [11, 15]). For instance, the Social Identity Model of Pro-Environmental Action (SIMPEA) [16] posits that individuals belonging to a social group tend to conform to the group's environmental commitments, while contrasting themselves with an outgroup diminishes their inclination to engage in pro-environmental behaviors [17, 18]. Several small-scale studies have hinted at underlying psychological factors such as perceived difficulty [11], personal norms [19], and national identity and attitudes [20, 21] influencing intervention effectiveness––but often focusing on a single intervention (e.g., [11, 17]). In addition to that, a recent meta-analysis has confirmed the positive role of psychological factors, such as attitudes, intentions, values, awareness, and emotions on energy-related choices (e.g., [22]). Yet, substantial gaps remain in the field concerning (i) the mediating and/or moderating roles of a large set of psychological factors, such as intentions to save electricity, perceived difficulty, attitudes to electricity saving, electricity saving habit strength, social norms, personal norms, collective efficacy, emotional reactions to electricity consumption, and national identity in the effectiveness of interventions within large-scale studies, and (ii) the variations in the magnitude of these mediating and/or moderating roles across different interventions. These limitations underscore the necessity for the application, combination, and comparison of a comprehensive array of existing programs within real-life, cost-effective, and pragmatic contexts [23, 24].

To address these issues, the present research protocol, as a part of the Energy efficiency through Behavior CHANge Transition (ENCHANT) project, is conducted based on a Randomized Controlled Trial (RCT), with the main aim of enhancing electricity saving through large-scale application of interventions (i.e., information provision, collective vs. individual message framing, social norms, consumption feedback, competition, and commitment strategies) targeting household electricity use behavior in six countries covering the geographical and cultural variability of Europe (i.e., Austria, Germany, Italy, Norway, Romania, and Türkiye). We seek to explore which (*combination* of) interventions are most effective in saving electricity among European households.

In the realm of RCTs, a comprehensive research protocol is essential. This protocol delineates the rationale, precise objectives, methodological intricacies, statistical analyses, and administrative details from the trial's inception to the final reporting of outcomes [25]. Ensuring robust internal and external validity is of paramount importance in RCTs [26]. Moreover, the dissemination of a RCT protocol serves as a mechanism through which environmental scholars can scrutinize the alignment of final analyses and outcomes with the researchers' original intentions [27].

## 1.2. Objectives

**1.2.1. Which intervention?.** The fundamental objective of the planned RCT is to enhance electricity saving behaviors, leading to reduced electricity consumption, and reduce carbon emissions in Europe. This is to be achieved through the strategic implementation of six carefully designed and rigorously controlled behavioral intervention strategies (i.e., information provision, collective vs. individual message framing, social norms, consumption feedback,

competitive elements, and commitment strategies). The primary research inquiry centers on the impact of these interventions, either single or combined, on the per capita electricity savings. By investigating the efficacy of various intervention patterns and their potential synergies, the study aims to elucidate the key determinants contributing to successful behavior change in the context of electricity saving.

**1.2.2. Through which psychological mechanisms?.** As the second objective, our study seeks to uncover the underlying psychological mechanisms, such as intentions to save electricity, perceived difficulty, attitudes to electricity saving, electricity saving habit strength, social norms, personal norms, collective efficacy, emotional reaction to electricity consumption, and national identity as the mediators and/or moderators of the effectiveness of interventions. S1 Table provides details regarding the hypotheses, considering main and interaction effects of independent on dependent variables in the RCT, along with the expected results.

## 2. Methods

The current trial protocol covers the recommended items presented by the Standard Protocol Items: Recommendations for Interventional Trials (SPIRIT) [25, 28]. See Fig 1 for the SPIRIT schedule and S1 Checklist for the SPIRIT checklist. This trial was preregistered in the Open Science Framework: https://osf.io/9vtn4

### 2.1. Trial design

The proposed web-based trial is implemented as a multicenter, double-blind, multiple-arm parallel randomized controlled trial across six countries. In the trial, there are 14 parallel groups, divided into 12 intervention groups and 2 control groups. Participants undergo assessments before, during, and after the interventions, resulting in a 14 (group) × 6 (time) mixed factorial design [refer to Figs 2 and 3 for a visual representation of ENCHANT's factorial design and recruitment stages, illustrating the allocation and intervention combination processes].

### 2.2. Study setting and participants

The comprehensive investigation of this web-based trial spans multiple European countries and regional contexts. The intended study is expected to recruit 1500 households in each of the six European countries, selected through a voluntary sampling method.

### 2.3. Eligibility criteria

Eligible participants should: (1) be aged 18 or above, (2) reside in Northern (Norway), Central (Austria, Germany), Eastern (Romania), or Southern Europe (Italy and Türkiye), (3) have access to an electricity meter for their household's consumption and pay for electricity based on actual consumption.

### 2.4. Recruitment, random assignment, and allocation

The ENCHANT project has different types of real-world user partners from three main actor categories: energy suppliers/manufacturers, local governments/governmental energy agencies, and energy-focused NGOs. The role of these user partners in the trial was to recruit participants for the interventions (see the participant flow chart [Fig 4]) through spreading the campaign message through diverse communication channels, including websites, posters, newsletters, social media platforms, messages in electricity bills, events, and newspaper advertisements. Representing various organizations and administrations across different countries such as Viken Fylkeskommune (NO), Naturvernforbundet (NO), Badenova (DE), Clusj

| STUDY PERIOD | | | | | | | | | |
|---|---|---|---|---|---|---|---|---|---|
| | Enrolment | Allocation | Post-allocation | | | | | | Close-out |
| TIMEPOINT | $-t_1$ | $0$ | $t_1$ | $t_2$ | $t_3$ | $t_4$ | $t_5$ | $t_6$ | $t_x$ |
| ENROLMENT: | X | | | | | | | | |
| Eligibility screen | X | | | | | | | | |
| Informed consent | X | | | | | | | | |
| Random assignment | X | | | | | | | | |
| Allocation | | X | | | | | | | |
| INTERVENTIONS: | | | | | | | | | |
| Control 1 | | | | | | | | | |
| Control 2 | | | | | | | | | |
| Intervention 1 | | | | ●———————● | | | | | |
| Intervention 2 | | | | ●———————● | | | | | |
| Intervention 3 | | | | ●———————● | | | | | |
| Intervention 4 | | | | ●———————● | | | | | |
| Intervention 5 | | | | ●———————● | | | | | |
| Intervention 6 | | | | ●———————————————● | | | | | |
| Intervention 7 | | | | ●———————————————● | | | | | |
| Intervention 8 | | | | ●———————————————● | | | | | |
| Intervention 9 | | | | ●———————————————● | | | | | |
| Intervention 10 | | | | ●———————● | | | | | |
| Intervention 11 | | | | ●———————————————● | | | | | |
| Intervention 12 | | | | ●———————● | | | | | |
| ASSESSMENTS: | | | | | | | | | |
| Demographic characteristics* | X | | | | | | | | |
| Primary outcomes | X | X | X | X | X | X | X | X | |
| Secondary outcomes | X | X | X | X | X | X | X | X | |

**Fig 1. Recommended content for the schedule of enrolment, interventions, and assessments.** *Notes.* * = including age, gender, household size, the number of individuals living in the household within distinct age brackets (under 6, 6–11, and 12–17), educational attainment, employment status, and perceived social status. More details are presented in Fig 3.

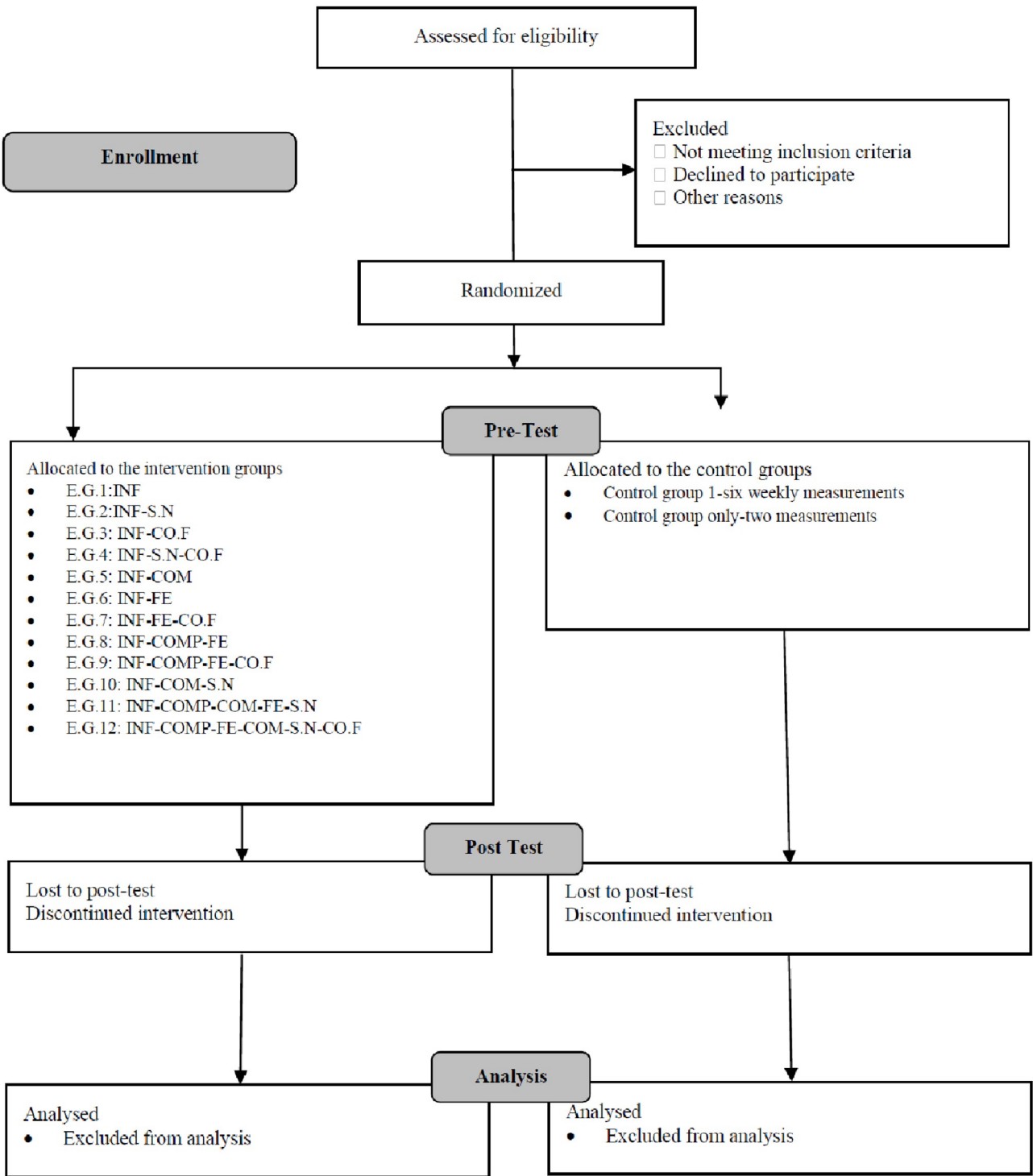

**Fig 2. Flow diagram of the progress through the phases of a 14-group parallel randomized trial flow.**

Napoca Municipality (RO), Electrica (RO), Izmir Metropolitan Municipality (TR), Gediz Elektrik (TR), EnergieKompass (AT), Energia Positiva (IT), and Fondazione Roffredo Caetani/Ninfa Gardens (IT), these user partners encourage individuals to register for the campaign on their respective country's landing page.

(A)

| | recruitment stage | W1 | W2 | W3 | W4 | W5 | W6 | |
|---|---|---|---|---|---|---|---|---|
| control group 1 | - | M | M | M | M | M | M | Electricity meter reading |
| weekly measurement | S1 | S2$_a$ | S2$_b$ | S2$_b$ | S2$_b$ | S2$_b$ | S2$_c$ | Surveys |
| control group 2 | - | M | - | - | - | - | M | Electricity meter reading |
| only two measurements | S1 | S2$_a$ | - | - | - | - | S2$_c$ | Surveys |
| Experimental group 1 | - | M | M | M | M | M | M | Electricity meter reading |
| information | - | - | X$_{info}$ | X$_{info}$ | X$_{info}$ | - | - | interventions |
| | S1 | S2$_a$ | S2$_b$ | S2$_b$ | S2$_b$ | S2$_b$ | S2$_c$ | Surveys |
| Experimental group 2 | - | M | M | M | M | M | M | Electricity meter reading |
| information | - | - | X$_{info}$ | X$_{info}$ | X$_{info}$ | - | - | interventions |
| social norm | - | - | - | X$_{SN}$ | X$_{SN}$ | - | - | interventions |
| | S1 | S2$_a$ | S2$_b$ | S2$_b$ | S2$_b$ | S2$_b$ | S2$_c$ | Surveys |
| Experimental group 3 | - | M | M | M | M | M | M | Electricity meter reading |
| information | - | - | X$_{info}$ | X$_{info}$ | X$_{info}$ | - | - | interventions |
| collective framing | - | - | X$_{coll}$ | X$_{coll}$ | X$_{coll}$ | - | - | interventions |
| | S1 | S2$_a$ | S2$_b$ | S2$_b$ | S2$_b$ | S2$_b$ | S2$_c$ | Surveys |
| Experimental group 4 | - | M | M | M | M | M | M | Electricity meter reading |
| information | - | - | X$_{info}$ | X$_{info}$ | X$_{info}$ | - | - | interventions |
| collective framing | - | - | X$_{coll}$ | X$_{coll}$ | X$_{coll}$ | - | - | interventions |
| social norm | - | - | - | X$_{SN}$ | X$_{SN}$ | - | - | interventions |
| | S1 | S2$_a$ | S2$_b$ | S2$_b$ | S2$_b$ | S2$_b$ | S2$_c$ | Surveys |
| Experimental group 5 | - | M | M | M | M | M | M | Electricity meter reading |
| information | - | - | X$_{info}$ | X$_{info}$ | X$_{info}$ | - | - | interventions |
| commitment | - | - | - | X$_{comm}$ | X$_{comm}$ | - | - | interventions |
| | S1 | S2$_a$ | S2$_b$ | S2$_b$ | S2$_b$ | S2$_b$ | S2$_c$ | Surveys |
| Experimental group 6 | - | M | M | M | M | M | M | Electricity meter reading |
| information | - | - | X$_{info}$ | X$_{info}$ | X$_{info}$ | - | - | interventions |
| feedback | - | - | - | - | X$_{feed}$ | X$_{feed}$ | - | interventions |
| | S1 | S2$_a$ | S2$_b$ | S2$_b$ | S2$_b$ | S2$_b$ | S2$_c$ | Surveys |
| Experimental group 7 | - | M | M | M | M | M | M | Electricity meter reading |
| information | - | - | X$_{info}$ | X$_{info}$ | X$_{info}$ | - | - | interventions |
| collective framing | - | - | X$_{coll}$ | X$_{coll}$ | X$_{coll}$ | - | - | interventions |
| feedback | - | - | - | - | X$_{feed}$ | X$_{feed}$ | - | interventions |
| | S1 | S2$_a$ | S2$_b$ | S2$_b$ | S2$_b$ | S2$_b$ | S2$_c$ | Surveys |

(B)

| | recruitment stage | W1 | W2 | W3 | W4 | W5 | W6 | |
|---|---|---|---|---|---|---|---|---|
| Experimental group 8 | - | M | M | M | M | M | M | Electricity meter reading |
| information | - | - | X$_{info}$ | X$_{info}$ | X$_{info}$ | - | - | interventions |
| competition | - | - | X$_{comp}$ | - | - | - | - | interventions |
| feedback | - | - | - | - | X$_{feed}$ | X$_{feed}$ | - | interventions |
| | S1 | S2$_a$ | S2$_b$ | S2$_b$ | S2$_b$ | S2$_b$ | S2$_c$ | Surveys |
| Experimental group 9 | - | M | M | M | M | M | M | Electricity meter reading |
| information | - | - | X$_{info}$ | X$_{info}$ | X$_{info}$ | - | - | interventions |
| collective framing | - | - | X$_{coll}$ | X$_{coll}$ | X$_{coll}$ | - | - | interventions |
| competition | - | - | X$_{comp}$ | - | - | - | - | interventions |
| feedback | - | - | - | - | X$_{feed}$ | X$_{feed}$ | - | interventions |
| | S1 | S2$_a$ | S2$_b$ | S2$_b$ | S2$_b$ | S2$_b$ | S2$_c$ | Surveys |
| Experimental group 10 | - | M | M | M | M | M | M | Electricity meter reading |
| information | - | - | X$_{info}$ | X$_{info}$ | X$_{info}$ | - | - | interventions |
| commitment | - | - | - | X$_{comm}$ | X$_{comm}$ | - | - | interventions |
| social norm | - | - | - | X$_{SN}$ | X$_{SN}$ | - | - | interventions |
| | S1 | S2$_a$ | S2$_b$ | S2$_b$ | S2$_b$ | S2$_b$ | S2$_c$ | Surveys |
| Experimental group 11 | - | M | M | M | M | M | M | Electricity meter reading |
| information | - | - | X$_{info}$ | X$_{info}$ | X$_{info}$ | - | - | interventions |
| competition | - | - | X$_{comp}$ | - | - | - | - | interventions |
| feedback | - | - | - | - | X$_{feed}$ | X$_{feed}$ | - | interventions |
| commitment | - | - | - | X$_{comm}$ | X$_{comm}$ | - | - | interventions |
| social norm | - | - | - | X$_{SN}$ | X$_{SN}$ | - | - | interventions |
| | S1 | S2$_a$ | S2$_b$ | S2$_b$ | S2$_b$ | S2$_b$ | S2$_c$ | Surveys |
| Experimental group 12 | - | M | M | M | M | M | M | Electricity meter reading |
| information | - | - | X$_{info}$ | X$_{info}$ | X$_{info}$ | - | - | interventions |
| collective framing | - | - | X$_{coll}$ | X$_{coll}$ | X$_{coll}$ | - | - | interventions |
| competition | - | - | X$_{comp}$ | - | - | - | - | interventions |
| feedback | - | - | - | - | X$_{feed}$ | X$_{feed}$ | - | interventions |
| commitment | - | - | - | X$_{comm}$ | X$_{comm}$ | - | - | interventions |
| social norm | - | - | - | X$_{SN}$ | X$_{SN}$ | - | - | interventions |
| | S1 | S2$_a$ | S2$_b$ | S2$_b$ | S2$_b$ | S2$_b$ | S2$_c$ | Surveys |

**Fig 3. ENCHANT platform experimental design.** *Notes*: There is a small deviation here for practical reasons: The control 2 group was informed that they are in a specific group that only provided measurements twice, not six times. This was done to make sure that they did not wonder if something was wrong when they were recruited for 6 times of measurement and then it goes silent for 4 weeks.

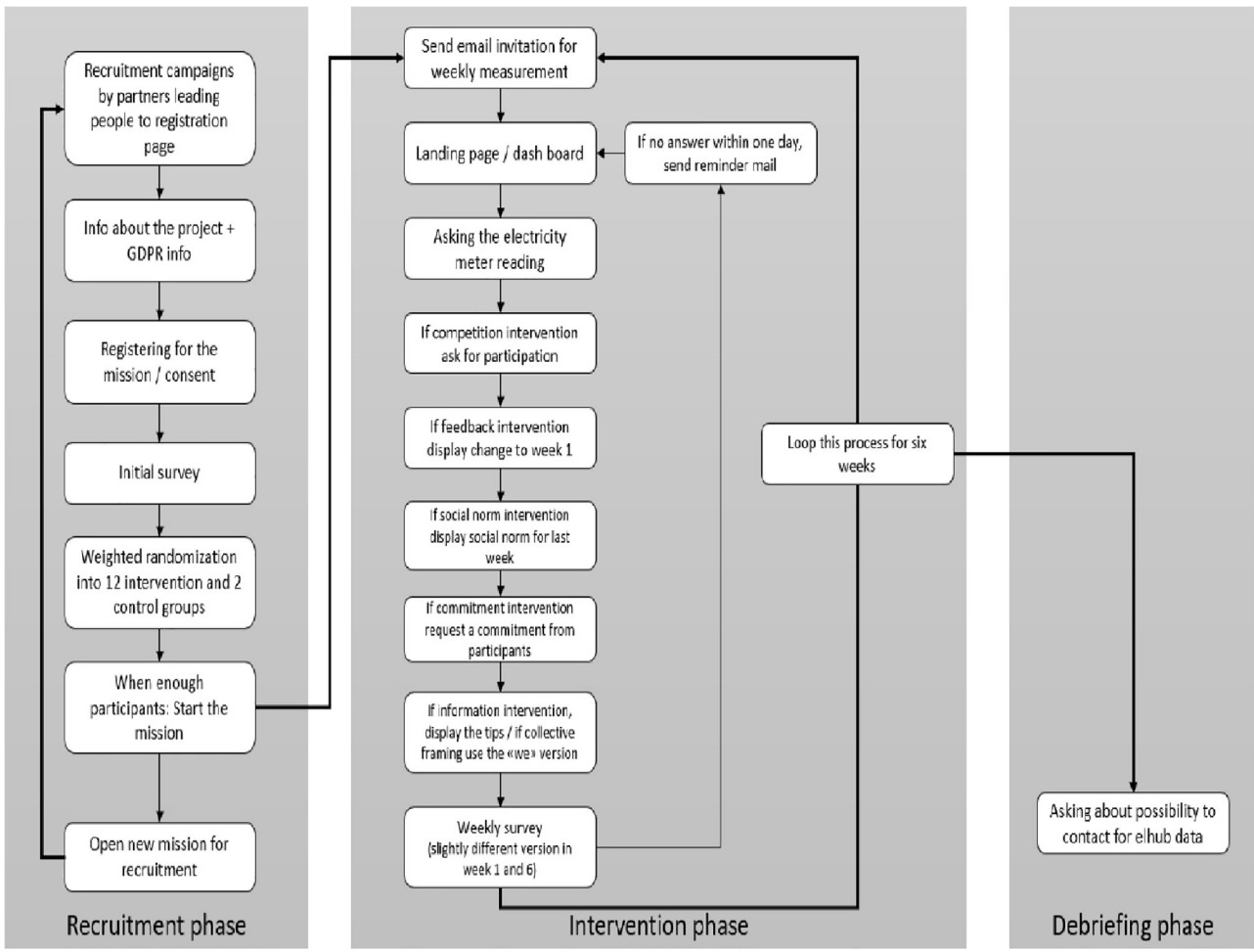

**Fig 4. ENCHANT platform: Participant flow chart.**

Registration of a participant for an intervention involves providing name and a valid e-mail address followed by the informed consent form along with a link to access the technical documentation of the General Data Protection Regulation (GDPR). Regular e-mails are sent to participants to invite them to provide weekly meter reading data and answer short surveys. Participants will be informed of the voluntary nature of participation and retain the right to withdraw their consent at any time without facing adverse consequences.

After registration, participants complete an initial survey encompassing demographic details, the current situation regarding larger electricity consuming devices (e.g., tumble dryers, charging of electric vehicles at home, and heating or cooling with electricity), and psychological factors influencing energy choices (e.g., perceived difficulty, electricity saving habit strength, personal norms, national identity). Subsequently, participants are randomly allocated to one of 14 groups, comprising 12 experimental and 2 control groups. The 6-week intervention begins once an adequate number of participants (n = 25) are recruited for each group so that participants do not have to wait for the start of the campaign too long. Simultaneously, recruitment efforts continue to form new groups that start later, ensuring an ample number of participants for each group [see Figs 2–4].

Recruitment priority was assigned to various groups, including both control groups, the information-only condition, all combinations of information paired with only one of the other interventions, as well as the full intervention packages presented in both individual and collective framing. We decided to prioritize some intervention combinations to enhance the statistical power for the most important analyses for the project (assessing the effects of each intervention on its own and the full package). The random assignment of eligible participants into intervention or control groups was automatically facilitated by a platform-generated random number list, with some groups receiving higher recruitment likelihoods due to our priority allocation strategy. Specifically, high-priority groups were assigned twice the recruitment likelihood compared to low-priority groups, and this allocation was managed within the platform.

## 2.5. Sample size

By September 2023, a total number of about 2000 participants have taken part and were included in the trial.

## 2.6. Socio-demographic characteristics

A range of demographic characteristics that have the potential to impact energy-related behaviors are collected, including age, gender, household size, the number of individuals living in the household within distinct age brackets (under 6, 6–11, and 12–17), educational attainment, employment status, and perceived social status.

## 2.7. Blinding

Given the nature of the present interventions, it is impractical to maintain blindness for both the researchers and the participants. The interventions remain undisclosed to the participants by not informing them about how other conditions looked like. Therefore, the trial assumes a double-blinded structure where the data analyst will be blinded to the interventions to minimize bias. To ensure this, an impartial data analyst or statistician will be responsible for conducting the data analysis without access to the intervention group codes.

## 2.8. Interventions

The proposed study will implement six distinct intervention types, which have been empirically established to influence behavior [6, 8, 29–37]. It is important to note that all tips (intervention messages) and survey questions regarding shifting energy away from peak periods (including checking the website or app of the energy provider) will only be presented in Norway (as smart meters are not less widely adopted in the other participating countries). Brief descriptions of these interventions are provided below:

**2.8.1. Informational provision.** The information intervention involves providing participants with relevant details, such as behaviors that lead to electricity savings. Various interventions include energy-saving tips and informational components [38, 39] and research emphasizes their influence on decisions related to energy consumption [40, 41].

In this trial, after an introductory text outlining the importance of one's contribution to electricity saving (e.g., "*You can do a lot of smaller and larger things that reduce your electricity consumption*"), we provide information in the form of a list of electricity saving tips in ten domains (plus one additional domain about load shifting in Norway, where most households have spot-price tariffs). An example from the domain "*cooking*" is the following: "*Be energy efficient when cooking: Cooking is a significant part of your electricity consumption. You can*

*save energy by heating water with a kettle instead of on the stove, only heating the amount of water you really need, keeping lids on pots, and keeping the oven door closed as much as possible*". These electricity saving tips were presented after all other interventions (if any) in weeks 2–4 of the six-week trial.

**2.8.2. Collective vs. individual framing.** Recognizing the importance of collective framing in energy-related decision-making [7], we aim to investigate and determine the most suitable framing strategies to maximize the impact of energy-related interventions. In the individual framing condition, the electricity saving tips are presented to individuals, while in the collective framing condition, the introductory text was rephrased to address for example "Norwegians" (or citizens of the other countries in their respective campaigns) as a social group (e.g., "*We in Norway can do a lot of smaller and larger things to reduce our electricity consumption*"). Then the individual tips were also addressing the social group instead of the individual as in the following example: "*Be energy efficient when cooking: Cooking is a significant part of our electricity consumption. We can save energy by heating water in a kettle instead of on the stove, only heating the amount of water we really need, keeping lids on pots, and keeping the oven door closed as much as possible.*" The collective framing informational provision was also presented in weeks 2–4 of the six-week trial.

**2.8.3. Social norms.** This intervention provides participants with information about the behavior of others and––through that––the socially accepted standards of conduct. Social norms feedback is a widely used intervention approach, which has been implemented in previous field experiments focused on energy conservation [8].

Our operationalization of the social norms' intervention was implemented as follows: the per person weekly consumption during last week was compared to the average consumption of the other participants in the same group during baseline. If the (normalized––see primary outcomes for more details) consumption was lower, the following message was displayed before listing the electricity saving tips: "*Congratulations! With xx kilowatt hours per person living in your household, you were below the average of people in the first week, which is xx kilowatt hours per person. Keep up the good work also next week. Maybe you can save even more electricity? Check out our tips!*" If the per person consumption during last week was above the average in the same group at baseline, the following message was displayed: "*With xx kilowatt hours per person living in your household, you were above the average of people in the first week, which is xx kilowatt hours per person. Maybe you find some of the electricity saving tips helpful to reduce the amount of electricity you use?*". The social norms intervention was presented in weeks 3 and 4 of the six-week trial.

**2.8.4. Consumption feedback.** The provision of feedback to participants about their historical and current electricity usage was used in the feedback intervention to reinforce and/or modify future actions [37]. In the context of the ENCHANT project, our primary aim is to explore the incorporation of feedback across various interventions.

In this intervention trial, the feedback intervention was provided before displaying the electricity consumption reduction tips, by displaying last week's electricity consumption in relation to the baseline week's consumption per person (roughly normalized for temperature effects for households heating with electricity). If the (normalized) consumption was lower than in the baseline week, the following text was presented: "*Bravo! You reduced your electricity consumption by xx% last week. Very well done. Keep up the good work. Maybe, you can save even more electricity by implementing some more of the tips we give you?*" If the (normalized) consumption was higher than in the baseline week, the following text was presented: "*Last week, your electricity consumption increased by xx%. Maybe this is a good motivation to try extra hard next week. Check out the saving tips for inspiration.*" Furthermore, the weekly

consumption was displayed as a graph on the personal dashboard. The feedback intervention was provided in weeks 4 and 5 of the six-week trial.

**2.8.5. Competition.** The competition intervention creates a scenario where participants engage in a contest, and those demonstrating the best performance receive a prize. The ENCHANT study also aims to pinpoint suitable target groups for competition-based energy conservation campaigns, since it. It acknowledges that competition might not be universally acceptable, particularly for individuals who may experience undue psychological pressure [42, 43].

In the trial, the competition intervention was implemented as follows. At the beginning of week 2, all participants in a competition condition were asked the following question: "*Saving electricity can be fun if you compete against other motivated people! We are setting up a competition about who is going to be the electricity saving champion. Would you like to be part of this competition and see how you are performing in relation to the best participants?*" Then they were given the following two options: (a) *Yes, I would like to participate*; (b) *No, I am not the competitive type*. If participants accepted the competition, their dashboard displayed a leaderboard of electricity savers within the group (approximately 20–25 participants who started together), indicating their relative position on the list. This was done while protecting participant privacy––while protecting participant privacy. The list could be sorted by absolute savings in kilowatt hours per person per week saved in relation to the baseline and by relative (percentage) saved.

**2.8.6. Commitment.** The commitment intervention requested participants to commit themselves to engage in future behaviors or specific behavioral goals of electricity consumption reduction and promoting sustainable practices. Commitment strategies have been utilized to incentivize sustainable behaviors, particularly in saving electricity and water [9, 44–46].

In the trial, declaration of commitment was collected two times, ones before week 3 and once before week 4 for all conditions including commitment. Specifically, participants were asked the following question: "*Making saving electricity part of your daily routines can be difficult and requires commitment. Research shows that it is easier, if you actively make a commitment to try hard and if you see that others do the same. Can we ask you to give us your commitment to make a real effort to save at least 5% of last week's electricity consumption in the coming week?*" Then the participants can choose between the following options: (a) *Yes, I commit that I will make a real effort to reduce my electricity consumption by at least 5% next week*; (b) *Yes, I am fine with that my commitment is listed for others to see to motivate them also to make an effort. Together, we can make a difference*; (c) *No, I do not want to commit to saving electricity next week*. Option (b) was only asked if people selected option (a). If option (b) was selected, the number of that participant was displayed in the dashboards of all other people in the group who already committed.

**2.8.7. Justification of the selection of conditions—Experimental setup.** In a full factorial design, implementing 2x2x2x2x2x2 would have resulted in a total of 64 different conditions. However, such an extensive design is not feasible, even in a large-scale field trial like the one implemented here. Furthermore, certain combinations of interventions are logically dependent on each other. For instance, competition inherently involves feedback about individual consumption, making it necessary to have an intervention in place for competition to occur. Considering these factors, we made the decision to reduce the number of conditions to 14. Among these, we prioritized 9 conditions. We achieved this by either increasing the sampling likelihood in countries where the number of recruited participants closely matched our target values or by excluding non-prioritized conditions in countries with lower recruitment numbers. Please see Table 1 for the justifications. The general rule for construction of the design was to prioritize the control conditions, the full packages with all interventions, and the single intervention

**Table 1. Justification for the choice and prioritization of intervention groups.**

| No | Condition | Priority | Justification |
|---|---|---|---|
| 1 | Control 1 | High | This control condition mimics the measurement regime of the intervention conditions, but does not provide any additional input to the participants. |
| 2 | Control 2 | High | As the weekly reading of the electricity meters might be an intervention in itself, we also implemented a control condition with only two measurements at the start and end of the whole five-week period. The five week consumption was then evenly distributed across the five weeks to calculate the average consumption for the analyses. |
| 3 | Information | High | Information about what to do is considered essential for changing Behavior. Therefore, information with electricity saving tips was presented in all intervention conditions. This condition tests the effect of information only. |
| 4 | Information + social norms | High | The main aim of this intervention group is to single out the effect that social norm communication can have on top of "just" providing information. |
| 5 | Information + collective framing | High | This is the collective version of the information only condition, testing if a collective framing of information given is changing the results. |
| 6 | Information + collective framing + social norm | Low | This is the collectively framed version of condition 4, which tests if the framing changes the effects of social norms. |
| 7 | Information + commitment | High | This condition tests if being asked for (and giving) commitment changes the effect of information given. |
| 8 | Information + feedback | Low | This condition tests if individual feedback on changes in electricity consumption changes the effect of information given. |
| 9 | Information + collective framing + feedback | Low | This is the collectively framed version of condition 8, which tests if the framing changes the effects of feedback. |
| 10 | Information + competition + feedback | High | This condition tests if a competitive element (and accepting this) changes the effect of information given and feedback without a competition. As a competition always requires giving feedback, there is no condition testing competition without feedback. |
| 11 | Information + collective framing + competition + feedback | Low | This is the collectively framed version of condition 10 which tests if the framing changes the effects of competition combined with feedback. |
| 12 | Information + commitment + social norm | Low | As social norms and commitment deal with the individual and social components of moral decision making, we also included a condition combining these two interventions (in addition to information as a basis). |
| 13 | Information + competition + feedback + commitment + social norm | High | This is the individually framed version of the whole intervention package. We assume that the combination of interventions is more effective than single interventions. |
| 14 | Information + competition + feedback + commitment + social norm + collective framing | High | This is the collectively framed version of the whole intervention package. We assume that the combination of interventions is more effective than single interventions. |

provisions. However, the information with the electricity saving tips was included with all interventions, as it was considered essential to provide some advice on how to save electricity. In addition, the collective framing was tested for all these combinations, though with lower priority for some combinations. Competition includes an element of feedback anyway, which is why competition is not tested individually but only in combination with feedback.

## 2.9. Measuring primary and secondary outcomes

Participants were asked to complete an initial survey lasting approximately 10 minutes, to monitor their electricity meter on a weekly basis, and to input the collected data into the ENCHANT's platform. Additionally, participants were asked to respond to a brief survey that took a maximum of 5 minutes each week. Detailed information about the experimental design parameters, components of the scales and items can be found in S2 Table and Fig 3. During the weekly phase, participants received email notifications informing them about the start of the next stage of the study, along with a link to access the dashboard. The dashboard displayed different information based on the specific experimental condition assigned to each participant. Within the dashboard, participants could perform three main tasks: (i) access

information about their electricity usage, (ii) submit their weekly measurements, and (iii) manage their profile (including ending their participation in the study or deleting their account). Details regarding independent and dependent variables, as well as socio-demographic and psychological covariates, along with moderator confounders, are available in S3 Table.

**2.9.1. Primary outcome.** *Electricity use*. Electricity use estimates were obtained through electricity meter readings and calculating the consumption based on the difference between readings. This encompassed one specific item: electricity consumption at 6 weeks (6-time points), measured as "consumption per person normalized to 7 days—kWh/week/person." Thus, differences in household size and variability in the measurement periods (if participants did not report in exactly 7-day intervals) were normalized before the analyses. In addition, electricity consumption was normalized by using data on temperature dependent heating and cooling needs (by extracting local heating and cooling degree days from the degreedays.net database) and electricity prices.

**2.9.2. Secondary outcomes.** To cover a broad spectrum of energy-related behaviors with significant potential for energy conservation, as outlined by Dietz, Gardner [47], a weekly self-report of implemented electricity saving behaviors was recorded. These behaviors encompassed investment, maintenance, and everyday curtailment behaviors. More specifically, investment behaviors in this study were measured as for example installing LED lights everywhere possible. An example for maintenance behavior promoted in the study is deicing the refrigerator or removing dust from its cooling coils. Examples of everyday curtailment behavior is included not using standby keeping lids on pots or washing laundry at lower temperatures (for more details, see S2 Table).

To build a comprehensive dataset of empirical impact data, capturing various contextual variables and outcomes from our targeted participants and users, we utilized multiple dimensions during the initial survey (S2 Table): (1) socio-demographics; (2) risk of energy poverty, (3) environmental concern, (4) personal norms, (5) ownership of electricity (in)efficient appliances, and (6) perceived behavioral control to save electricity. Additionally, during the weekly evaluations at the 6 different time points, we measured the following components (S2 Table): (1) peak hour consumption (only in Norway), (2) intentions to save electricity during the following week; (3) attitudes to save electricity, (4) perceived Behavior al control, (5) perceived difficulty of implementing the different energy tips, (6) emotional reaction to electricity consumption, (7) social norms, (8) collective efficacy, and (9) self-reported implementation of electricity saving behaviors. ENCHANT's comprehensive measurement approach ensures the robustness and reliability of the data collected, enhancing the validity of the project's findings.

**2.9.3. Feasibility and acceptability measures.** The study aims to evaluate the practicality and willingness to participate in web-based intervention randomized controlled trials (RCTs) with households. This approach is consistent with recommendations found in the literature regarding the assessment of client/participants satisfaction and the evaluation of user satisfaction with mobile health app [48, 49]. This assessment will encompass dropout, engagement, and response rate calculations. Moreover, the acceptability of web-based interventions concerning energy-related behaviors will be explored by asking the participants after completion of the study, how far they experienced the study as an encroachment of their personal freedom. These measures will offer insights into the practicality and acceptability of web-based interventions, providing a comprehensive understanding of participants' engagement and perceptions. Furthermore, we will analyze which channels were most successful in recruiting participants (also taking their successive dropouts and electricity savings into account).

## 2.10. Data analysis and interpretation methods

To ensure a robust evaluation of the interventions' impact on consumers' energy efficiency behavior, ENCHANT employs a comprehensive data analysis strategy. Embracing the intent-to-treat (ITT) principle, all data collected from randomized participants will be incorporated into the analyses. The analysis process involves meticulous steps including data screening, consistency checks, descriptive and graphical analyses, and outlier detection. Decisions concerning outlier removal will be based on comparisons of means and 5% trimmed means. Thus, both original data and robust estimation methods will contribute to the analysis.

Addressing missing data is pivotal, and imputation techniques will be implemented to handle such instances. Intent-to-treat analyses will be conducted to evaluate the impact of missing data on outcome variables, with the anticipation that it will not significantly affect the main study findings. Hence, imputed missing data will drive subsequent analyses. Additionally, participants' demographic characteristics will undergo a thorough examination, with Chi-square tests and independent t-tests utilized to assess randomization effectiveness and to identify potential variations between intervention and control groups. Ensuring dataset pairing will facilitate accurate paired comparisons across groups.

Mixed Regression Analysis (MRA) will be deployed, encompassing both within-subject and between-subject factors. The within-subject factor will encompass assessment time points, including pre-intervention and post-intervention, with 4 additional measurement time points in between. This results in a total of 6 measurements for both psychological variables and electricity consumption readings, covering a period of 5 weeks. Furthermore, demographic variables known to influence outcomes (e.g., gender and education) will be considered as covariates. Beyond the main analysis, the project will employ the Reliable Change Index (RCI) scores based on the Jacobson and Truax [50] Index––to gauge the magnitude of change and explore practically significant differences in electricity savings and renewable energy adoption pre- and post-interventions. Electricity consumption raw data will be normalized to 7-day per person rates adjusted for temperature effects by using Heating Degree Days (HDD) and Cooling Degree Days (CDD) with 15 Degrees Celsius as cut-off point as covariates (see degreedays. net for a background of the calculation and use of HDD and CDD). Additionally, average weekly electricity prices will be used as covariates to control for effects of varying electricity prices. The effects of HDD, CDD, and electricity prices depend on the building structures and heating/cooling technologies as well as household electricity pricing schemes. Therefore, the interactions between countries and these factors as well as between heating/cooling technologies used by each individual household and these factors will be included in the analyses Information on data management and confidentiality can be found in S1 File.

Moderation effects will be tested through interaction terms, and our analysis proceeds in three steps: (1) analysis of intervention effects, (2) covariate adjustments, and (3) inclusion of interactions. To explore mediation effects, we employ a multilevel Structural Equation Modeling (SEM) approach, nested within the five-week data collection period. This approach evaluates the roles of person-specific and time-specific factors in predicting weekly electricity use. Our statistical strategy allows us to thoroughly examine the impact of interventions, considering various potential moderators and mediators, and controlling for important covariates. This approach enhances the robustness of our findings and helps uncover the complex dynamics of electricity conservation behavior change.

Our approach to addressing multiplicity in the analysis and interpretation of both primary and secondary outcomes varies based on the number of tests conducted. We may employ either the False Discovery Rate (FDR) approach or the Holm-Bonferroni method for correction, selecting the most appropriate method for the situation at hand. The choice aligns with

any preregistered methods, and we ensure that alpha in power calculations is adjusted accordingly to match the chosen correction method.

## 2.11. Research ethics approval

The ENCHANT's research and user partners have established robust operational policies that effectively tackle ethical concerns, ensuring unwavering compliance with European and national regulations, as well as adhering to professional codes safeguarding personal data. This commitment extends to directives such as 95/46/EC and GDPR, underscoring the project's dedication to data protection. ENCHANT's inclusive composition encompasses both EU and non-EU beneficiaries, ensuring that legal data transfers are executed in accordance with established protocols. The trial has been approved by the Norwegian Agency for Shared Services in Education and Research (SIKT, formerly NSD; case number 120694) and by the data protection officer of the Norwegian University of Science and Technology.

## 2.12. Consent or assent

Informed consent forms will be made available to all participants engaged in the ENCHANT project. A pivotal feature of the project is the participants' autonomy to retract their consent at any juncture of the campaign. This can be easily executed by either self-administering their campaign participation or opting to delete their account, thereby effecting the comprehensive removal of their data from the system. Additionally, participants retain the option to have their data manually erased by directly contacting the designated responsible individual, as specified in the consent form (see S2 File). Should participants desire access to their stored data, a straightforward recourse is available: they can reach out to the principal responsible person overseeing the study and promptly obtain a copy of all data pertinent to their involvement. The full version of our study protocol that received ethics approval can be seen in S2 File.

## 3. Results

The data collection phase for the RCT was initiated in January 2023 and is expected to extend until the culmination of our endeavors in October 2023. In the transition to September 2023, about 2000 dedicated participants were already successfully recruited into the study. It is important to note that, as of this juncture, not all participants have completed all aspects of the protocol. This stage is important as a foundational cornerstone toward achieving our ultimate objective of deriving insightful outcomes and deepening our comprehension within the ENCHANT project. The project is scheduled to conclude on December 31, 2023.

## 4. Discussion

The evaluation of intervention strategies rooted in psychological and behavioral sciences has been a topic of exploration in various field trials aimed at improving electricity savings. These trials have provided valuable insights into the factors that influence effective decision-making, as well as the challenges and contextual variations impacting energy efficiency choices within households [2–5]. Building upon these earlier findings, the ENCHANT project takes a significant stride with this RCT by implementing well-established, evidence-based behavioral interventions on a large scale. Through the engagement of a cohort of European citizens across six countries, the proposed trial undertakes a large-scale RCT, to assess the effectiveness of intervention strategies that promote electricity saving among households throughout Europe.

Observations in natural settings may help to establish robust causal claims, important for evidence-driven policymaking. Nevertheless, many behavioral interventions, when

implemented in practical scenarios, tend to lose their efficacy in promoting energy efficiency in real-life contexts [7–9], addressing the need for more large-scale real-time interventions in this field. The present study provides a trial protocol for real-time interventions, aiming to promote electricity savings. Utilizing the RCT methodology, the project furnishes empirical evidence across diverse contexts, useful for informing evidence-based policies. It has been suggested that the modest impact of behavioral interventions on electricity saving might stem from the presence of psychological mediators and/or moderators that have been largely overlooked. Indeed, individuals' psychological characteristics can either facilitate or impede the desired outcomes of interventions (e.g., [11, 15, 20, 21]). The intended large-scale RCT will take into consideration the probable impact of intentions to save electricity, perceived difficulty, attitudes to electricity saving, electricity saving habit strength, descriptive and injunctive social norms, personal norms, collective efficacy, emotional response to electricity consumption, and national identity in the effectiveness of energy use interventions. By assessing the possible role of psychological factors on the efficacy of the proposed RCTs (either contributing or confounding), this study offers valuable insights for evidence-based strategies, addressing existing research limitations.

### 4.1. Limitations

A potential challenge of the proposed RCT pertains to the design and execution of the interventions. While RCTs offer a robust methodology for establishing causal relationships between interventions and outcomes, they are not without their complexities. In the context of ENCHANT, the intricate nature of behavioral interventions, especially across diverse cultural and contextual settings, introduces potential hurdles in implementing a standardized RCT approach. Variations in local conditions and participant responses may introduce variability in intervention outcomes, thereby complicating results' interpretation. Furthermore, the fact that the included countries are in different climate zones may further complicate the interpretation. On the most basic level, cultural differences might even result in strongly varying success of the recruitment to the RCT. Furthermore, conducting a large-scale RCT necessitates meticulous coordination, substantial resources, and thorough consideration of potential biases. Despite ENCHANT's comprehensive planning, unforeseen variables during the trial's progression could impact the validity and generalizability of the obtained findings. It is worth noting that RCTs offer a snapshot of a specific timeframe, potentially making it challenging to fully capture longitudinal effects and the sustainability of behavioral changes within the project's defined timeline. Finally, although this planned trial is a large-scale field experiment that includes heterogeneous samples from broad demographics, covering all population segments from each country was beyond the scope of this trial and can be addressed in future research.

To mitigate these limitations, a proactive strategy is imperative. This involves continuously adapting intervention strategies based on real-time feedback and meticulously considering potential biases and contextual nuances. In light of this, collaboration among behavioral experts, project partners, and stakeholders will play a pivotal role in enhancing the effectiveness and reliability of interventions within the ENCHANT project.

## 5. Conclusion

The outcomes of the intended RCT are poised to facilitate electricity savings and to advance our knowledge in the fields of energy behavior change and environmental psychology. Through a comprehensive evaluation of six types of interventions and their impacts on varied populations, this research initiative will empower businesses and policymakers to enact precise and impactful interventions that foster electricity savings. The insights garnered from the

results of our trial across six European countries may assist decision-makers in formulating well-informed policy strategies for a more ecologically responsible future. This may occur when taking into consideration people's psychological characteristics as mediators or moderators of the interventions that are included in the present trial. Ultimately, the intended trial may illuminate the efficacy of interventions, thereby steering us closer to a more sustainable world.

## Supporting information

**S1 Table. Hypotheses considering main and interaction effects of independent and dependent variables in the RCT and expected results.**
(DOCX)

**S2 Table. Items pertaining to electricity-related behaviors/values/habits with substantial potential for energy conservation.**
(DOCX)

**S3 Table. Independent and dependent variables, socio-demographic and psychological covariates, and moderator confounders.**
(DOCX)

**S1 Checklist. SPIRIT checklist.**
(DOC)

**S1 File. Data management and confidentiality.**
(DOCX)

**S2 File. Are you interested in taking part in the research project "Energy Efficiency in Norwegian Households"?**
(DOCX)

**S3 File. Study protocol approved by ethics committee.**
(PDF)

## Acknowledgments

The authors gratefully thank those who kindly participated in this research.

## Author Contributions

**Conceptualization:** Mojtaba Habibi Asgarabad, Stepan Vesely, Mehmet Efe Biresselioglu, Giuseppe Carrus, Muhittin Hakan Demir, Andrea Kollmann, Christian A. Klöckner.

**Data curation:** Christian A. Klöckner.

**Formal analysis:** Christian A. Klöckner.

**Funding acquisition:** Mehmet Efe Biresselioglu, Giuseppe Carrus, Muhittin Hakan Demir, Andrea Kollmann, Christian A. Klöckner.

**Investigation:** Mojtaba Habibi Asgarabad, Christian A. Klöckner.

**Methodology:** Mojtaba Habibi Asgarabad, Christian A. Klöckner.

**Project administration:** Christian A. Klöckner.

**Resources:** Christian A. Klöckner.

**Software:** Christian A. Klöckner.

**Supervision:** Christian A. Klöckner.

**Validation:** Mojtaba Habibi Asgarabad, Stepan Vesely, Christian A. Klöckner.

**Visualization:** Mojtaba Habibi Asgarabad, Christian A. Klöckner.

**Writing – original draft:** Mojtaba Habibi Asgarabad.

**Writing – review & editing:** Mojtaba Habibi Asgarabad, Stepan Vesely, Mehmet Efe Biresselioglu, Federica Caffaro, Giuseppe Carrus, Muhittin Hakan Demir, Benjamin Kirchler, Andrea Kollmann, Chiara Massullo, Lorenza Tiberio, Christian A. Klöckner.

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
