## [Editor Report · Decision Letter 0]

16 Oct 2023

Promoting Electricity Conservation through Behavior Change: A Study Protocol for a Web-Based Multiple-Arm Parallel Randomized Controlled Trial

PONE-D-23-32320

Dear Dr. Habibi Asagarabad,

We’re pleased to inform you that your manuscript has been judged scientifically suitable for publication and will be formally accepted for publication once it meets all outstanding technical requirements.

Kind regards,

Avanti Dey, PhD

Staff Editor

PLOS ONE

 [The ENCHANT project is funded by European Union’s Horizon 2020 research and innovation programme [project number 957115].].  

Please state what role the funders took in the study.  If the funders had no role, please state: ""The funders had no role in study design, data collection and analysis, decision to publish, or preparation of the manuscript.

Please respond by return e-mail so that we can amend your financial disclosure and competing interests on your behalf.

5. PLOS requires an ORCID iD for the corresponding author in Editorial Manager on papers submitted after December 6th, 2016. Please ensure that you have an ORCID iD and that it is validated in Editorial Manager. To do this, go to ‘Update my Information’ (in the upper left-hand corner of the main menu), and click on the Fetch/Validate link next to the ORCID field. This will take you to the ORCID site and allow you to create a new iD or authenticate a pre-existing iD in Editorial Manager. Please see the following video for instructions on linking an ORCID iD to your Editorial Manager account: ". " ext-link-type="uri" xlink:type="simple">https://www.youtube.com/watch?v=_xcclfuvtxQ"". 

6. We note that Figure 2Sa, 2Sb, 3Sa, 3Sb, 4Sa, 4Sb, 5S and 6S  in your submission contain copyrighted images. All PLOS content is published under the Creative Commons Attribution License (CC BY 4.0), which means that the manuscript, images, and Supporting Information files will be freely available online, and any third party is permitted to access, download, copy, distribute, and use these materials in any way, even commercially, with proper attribution. For more information, see our copyright guidelines: http://journals.plos.org/plosone/s/licenses-and-copyright.

A. You may seek permission from the original copyright holder of Figure 2Sa, 2Sb, 3Sa, 3Sb, 4Sa, 4Sb, 5S and 6S to publish the content specifically under the CC BY 4.0 license. 

B. If you are unable to obtain permission from the original copyright holder to publish these figures under the CC BY 4.0 license or if the copyright holder’s requirements are incompatible with the CC BY 4.0 license, please either i) remove the figure or ii) supply a replacement figure that complies with the CC BY 4.0 license. Please check copyright information on all replacement figures and update the figure caption with source information. If applicable, please specify in the figure caption text when a figure is similar but not identical to the original image and is therefore for illustrative purposes only.

Additional Editor Comments (optional):

We have deemed your submission eligible for expedited acceptance based on the documents provided, and the quality of your work. Thank you for submitting your work to PLOS ONE!
---

## [Editor Report · Acceptance letter]

25 Jan 2024

PONE-D-23-32320 

PLOS ONE

Dear Dr. Habibi Asgarabad, 

I'm pleased to inform you that your manuscript has been deemed suitable for publication in PLOS ONE. Congratulations! Your manuscript is now being handed over to our production team.

Kind regards, 

on behalf of

Dr. Avanti Dey 

Staff Editor

PLOS ONE